# Goal-Aware Cross-Entropy
# for Multi-Target Reinforcement Learning

**Kibeom Kim**[1,2], **Min Whoo Lee**[1], **Yoonsung Kim**[1], **Je-Hwan Ryu**[1],
**Minsu Lee**[1,3]*, **Byoung-Tak Zhang**[1,3]*
[1]Seoul National University, [2]Surromind, [3]AIIS[†]
{kbkim, mwlee, yskim, jhryu, mslee, btzhang}@bi.snu.ac.kr

## Abstract

Learning in a multi-target environment without prior knowledge about the targets requires a large amount of samples and makes generalization difficult. To solve this problem, it is important to be able to discriminate targets through semantic understanding. In this paper, we propose goal-aware cross-entropy (GACE) loss, that can be utilized in a self-supervised way using auto-labeled goal states alongside reinforcement learning. Based on the loss, we then devise goal-discriminative attention networks (GDAN) which utilize the goal-relevant information to focus on the given instruction. We evaluate the proposed methods on visual navigation and robot arm manipulation tasks with multi-target environments and show that GDAN outperforms the state-of-the-art methods in terms of task success ratio, sample efficiency, and generalization. Additionally, qualitative analyses demonstrate that our proposed method can help the agent become aware of and focus on the given instruction clearly, promoting goal-directed behavior.

## 1 Introduction

Reinforcement learning (RL) has been expanding to various fields including robotics, to solve increasingly complex problems. For instance, RL has been gradually mastering skills such as robot arm/hand manipulation on an object [2, 48, 36, 22] and navigation to a target destination [18, 39]. However, to benefit humans like the R2-D2 robot in the Star Wars, RL must extend to realistic settings that require interaction with multiple objects or destinations, which is still challenging for RL.

For this reason, it is important for multi-target tasks to be considered. We use the term *multi-target tasks* to refer to tasks that require the agent to interact with variable goals. In a multi-target task, *targets* are possible goal candidates, which may be objects or key entities that play a decisive role in determining the success or failure of the task execution. The *goals* may be selected among the targets by the current multi-target task, specified with a cue or an instruction such as "Bring me a {*spoon, cup* or *specific object*}" or "Go to the {*kitchen, livingroom* or *specific destination*}". The states in which the agent reaches the goal are called *goal states*.

Reinforcement learning allows learning multi-target or instruction-based tasks in an end-to-end manner. Latest reinforcement learning studies on these tasks mainly focus on prior knowledge about targets [27, 30, 8]. Other studies focus on learning representation on the environment [46] or about the targets only implicitly [44]. These methods lead to insufficient understanding of the goal, and sample-inefficiency and generalization problems arise.

For this matter, we propose a *Goal-Aware Cross-Entropy (GACE)* loss and *Goal-Discriminative Attention Networks (GDAN)*[2] for multi-target tasks in reinforcement learning. These methods, unlike

---

*Corresponding authors. [†]AI Institute, Seoul National University

[2]Code available at `https://github.com/kibeomKim/GACE-GDAN`

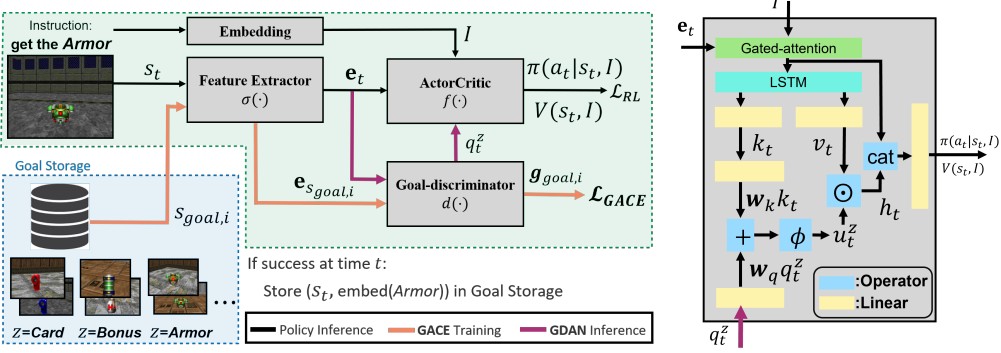

(a) Architecture from Goal-Aware Cross-Entropy and GDAN       (b) Attention in ActorCritic

Figure 1: Overview of the proposed architecture. (a) A goal-aware visual representation learning method by $\mathcal{L}_{GACE}$ from goal-aware cross-entropy loss. The loss updates the feature extractor and the goal-discriminator. (b) In GDAN, the ActorCritic utilizes the information from the goal-discriminator as the query for goal-directed actions. See the details in Section 3.3 and 3.4.

previous studies without prior knowledge, allow semantic understanding of goals including their appearances and other characteristics. The agent automatically labels and collects goal states data through trial-and-error in a self-supervised manner. Based on these self-collected data, we use GACE loss as an auxiliary loss to train a goal-discriminator that learns goal representations. Lastly, the GDAN extracts the information from the goal-discriminator as a goal-relevant query, with which an attention is performed to infer goal-directed actions.

We additionally propose visual navigation and robot arm manipulation tasks as benchmarks for multi-target task experiments. These tasks involve targets which are the multiple types of objects randomly placed within the environments. In the benchmarks, we make these tasks visually complex using randomized background textures, allowing us to evaluate generalization.

In summary, the contributions of this paper are as follows:

- We propose a Goal-Aware Cross-Entropy loss for learning auto-labeled goal states in a self-supervised manner, solving instruction-based multi-target tasks in reinforcement learning.

- We additionally propose Goal-Discriminative Attention Networks that use the goal-relevant query from the goal-discriminator to focus exclusively on important goal-related information.

- Our method achieves significantly better performances in terms of the success ratio, sample-efficiency, and generalization on visual navigation and robot arm manipulation multi-target tasks. In particular, compared with the baseline methods, our method excels in Sample Efficiency Improvement metric by **17 times** in visual navigation task and by **4.8** times in manipulation task.

- We present two instruction-based benchmarks for goal-based multi-target environments, to interact with multiple targets for realistic settings. These benchmarks are made publicly available, along with the implementation of our proposed method.

## 2 Related work

**Multi-target tasks in reinforcement learning**    There has been a sparsity of studies that aim to solve multi-target tasks with reinforcement learning. Wu et al. [44] proposes instruction-based indoor environments, as well as attention networks for multi-target learning to respond to the given instruction. Deisenroth and Fox [11] uses a model-based policy search method for multiple targets, instead of learning goal-relevant information. The aforementioned methods are for learning dynamics model or learning targets indirectly for multiple target tasks. In our method, on the other hand, the policy directly learns to distinguish goals without the dynamics model.

The terms *targets* and *goals* have been used in diverse manners in the existing literature. There are subgoal generation methods that generate intermediate goals such as imagined goal [31, 14] or random goals [34] to help agents solve the tasks. Multi-goal RL [47, 35, 13, 10] aims to deal with multiple tasks, learning to reach different goal states for each task. Scene-driven visual navigation tasks [49, 12, 30] specify the goal by an image. However, these tasks require the agent only to search for visually similar locations or objects, rather than gaining semantic understanding of goals and promoting generalization. Instruction-based tasks [1, 40] focus on learning to follow detailed instructions. These tasks have a different purpose from our tasks, where we focus on having appropriate interactions with multiple targets depending on the instruction.

**Representation learning in reinforcement learning** To improve performance in RL, several recent works have used a variety of representation learning methods. Among these methods, many make use of auxiliary tasks taught via unsupervised learning techniques such as VAE [24, 19] for learning dynamics [16, 45, 38] and contrastive learning method [7] for learning representations [26] of a given environment. Nair et al. [32] uses goal states, specified as the last states of the trajectories, to learn the difference between future state and goal state for redistributing model error in model-based reinforcement learning. The aforementioned methods are for learning representations of the environment rather than for learning those of key states for solving the task. In contrast, our method directly learns goal-relevant state representations.

Jaderberg et al. [21] proposes various auxiliary tasks for representation learning in RL. One such method is reward prediction, which learns the environment by predicting the reward of the future state from three classes: {positive, 0, negative}. This method focuses on myopic reward prediction, and not on ultimately learning the goal state to solve the task. Consequently, it is not a suitable method to apply to multi-target environments, where the goal can be selected among a diverse range of objects.

**Attention methods in reinforcement learning** The attention method was initially introduced for natural language processing [3] but is now being actively studied in various fields such as computer vision [43]. There are also attention-based methods for reinforcement learning which use inputs from different parts of the state [4, 9]. Other methods include the use of encoding information of sequential frames up to the last step [29], as well as the application of self-attention [6]. Unlike these methods, our approach explicitly extracts and focuses solely on goal-related information for solving a task.

## 3 Method

### 3.1 Preliminary

Reinforcement learning (RL) from Sutton and Barto [41] aims to maximize cumulative rewards by trial-and-error in a Markov Decision Process (MDP). An MDP is defined by a tuple $(\mathcal{S}, \mathcal{A}, \mathcal{R}, \mathcal{P}, \gamma)$, where $\mathcal{S}$ is the set of states, $\mathcal{A}$ is the set of actions, $\mathcal{R} : \mathcal{S} \times \mathcal{A} \to \mathbb{R}$ is the reward function, $\mathcal{P} : \mathcal{S} \times \mathcal{A} \times \mathcal{S} \to \mathbb{R}$ is the transition probability distribution, and $\gamma \in (0,1]$ is the discount factor. At each time step $t$, the agent observes state $s_t \in \mathcal{S}$, selects an action $a_t \in \mathcal{A}$ according to its policy $\pi : \mathcal{S} \to \mathcal{A}$, and receives a reward $r_t$ and next state $s_{t+1}$. In finite-horizon MDPs, return $R_t = \Sigma_{k=0}^{T-t} \gamma^k r_{t+k}$ is accumulated discounted rewards, where $T$ is the maximum episode length. State value function $V(s) = \mathbb{E}[R_t | s_t = s]$ is the expected return from state $s$.

**Instruction-based multi-target reinforcement learning** Universal value function approximators [37] estimate state value function as $V(s, x)$, for jointly learning the task from embedded state $s$ and goal information $x$. This is relevant to the notions of multi-target RL that we define as below.

Multi-target environments contain $N$ *targets*, or goal candidates $\mathcal{T} = \{z_1, z_2, ..., z_N\}$. These environments provide an instruction $I^z$ that specifies which target the agent must interact with, or the *goal* $z \in \mathcal{T}$. The instruction is given randomly every episode, in the form such as "Get the *Bonus*" or "Reach the *Green Box*" as in our benchmark tasks, in which cases the goals are *Bonus* and *Green Box* respectively. Hence, the state value function in such task is $V(s, I^z) = \mathbb{E}[R_t | \bar{s}_t = (s, z)]$, where $\bar{s}_t$ is state conditioned on the goal $z$. The policy $\pi$ is also conditioned on the instruction $I^z$ as $\pi(a_t | s_t, I^z)$ where $a_t$ is action at time $t$ given state $s_t$ and $I^z$.

## 3.2 Auto-labeled goal states for self-supervised learning

Prior to describing the main method in Sec. 3.3 and 3.4, this section explains the automatic collection of goal data for self-supervised learning. Suppose that the instruction $I^z$ specifies the target $z$ as the goal and the episode ends when the goal is reached at time step $t'$. We refer to the state $s_{t'}$ as a *goal state*, which we assume is highly correlated with the reward or the rewarding state. Throughout the training, the reached goal label $z$ and the corresponding goal state $s_{t'}$ are automatically collected as a tuple $(s_{t'}, one\_hot(z))$, called *storage data*. Rather than being manually provided the goal information, the agent actively gathers the data needed to learn the goals, relying only on the instruction $I^z$ given by the environment. This allows the agent to learn in an end-to-end, self-supervised manner. The states for the failed episodes are also stored as a negative case with low probability $\epsilon_N$. We clarify that the storage data does not serve as the prior during the learning and is instead used for training alongside the multi-target reinforcement learning task.

## 3.3 Goal-discriminator by goal-aware cross-entropy

Our proposed method adds an auxiliary task to the main reinforcement learning task as shown in Figure 1a. A general reinforcement learning model consists of a feature extractor $\sigma(\cdot)$, which converts state $s_t$ to an encoding $\mathbf{e}_t$, and an ActorCritic $f(\cdot)$ that outputs a policy $\pi$ and value $V$:

$$\mathbf{e}_t = \sigma(s_t) \tag{1}$$
$$\pi(a_t|s_t, I), V(s_t, I) = f(\mathbf{e}_t, I) \tag{2}$$

where $I$ is the instruction. The details of the ActorCritic $f(\cdot)$ and the resulting loss $\mathcal{L}_{RL}$ depend on the specific reinforcement learning algorithm that is chosen. In our visual navigation experiments, we use asynchronous advantage actor-critic (A3C) [28] as the main algorithm, where the loss $\mathcal{L}_{RL}$ is defined as the following

$$\mathcal{L}_p = \nabla \log \pi(a_t|s_t, I)(R_t - V(s_t, I)) + \beta \nabla H(\pi(a_t|s_t, I)) \tag{3}$$
$$\mathcal{L}_v = (R_t - V(s_t, I))^2 \tag{4}$$
$$\mathcal{L}_{RL} := \mathcal{L}_{A3C} = \mathcal{L}_p + 0.5 \cdot \mathcal{L}_v \tag{5}$$

where $\mathcal{L}_p$ and $\mathcal{L}_v$ respectively denote policy and value loss, $R_t$ denotes the sum of decayed rewards from time steps $t$ to $T$, and $H$ and $\beta$ denote the entropy term and its coefficient respectively. See Appendix D for algorithm details.

The vanilla base algorithm is inefficient and indirect at gaining an understanding of goals. Suppose that we have a set of goal states $\mathcal{S}_z \subset \mathcal{S}$ corresponding to each goal $z$. Intuitively, given an instruction $I^z$ that specifies $z$, the agent must learn the value function $V(s, I)$ that discriminates the goals, such that $V(s_{goal}^z, I^z) > V(s_{goal}^j, I^z)$ for $s_{goal}^z \in \mathcal{S}_z$ and $s_{goal}^j \in \mathcal{S}_j, \forall z \neq j$.

To achieve this effect, we propose *Goal-Aware Cross-Entropy* (GACE) loss as our contribution, which trains the goal-discriminator that facilitates semantic understanding of goals alongside the policy in Figure 1a. The inference process is as follows. First, the storage data collected in Sec. 3.2 are sampled as a batch of $s_{goal,i}$'s (with $i$ as data index within the batch) and fed into the feature extractor $\sigma(\cdot)$ for state encoding in Eq. 6. Next, the encoded input data $\mathbf{e}_{s_{goal,i}}$ is fed into the goal-discriminator $d(\cdot)$, a multi-layer perceptron (MLP), in Eq. 7. This yields a prediction $\mathbf{g}_{goal,i}$, a vector that contains probability distribution over possible goals that $s_{goal,i}$ belongs to.

$$\mathbf{e}_{s_{goal,i}} = \sigma(s_{goal,i}) \tag{6}$$
$$\mathbf{g}_{goal,i} = d(\mathbf{e}_{s_{goal,i}}) \tag{7}$$

From the output $\mathbf{g}_{goal,i}$, we calculate the Goal-Aware Cross-Entropy loss $\mathcal{L}_{GACE}$ as Eq. 8, where $M$ is the batch size, and $z_i$ is the automatic label corresponding to state $s_{goal,i}$.

$$\mathcal{L}_{GACE} = -\sum_{i=0}^{M-1} one\_hot(z_i) \cdot \log(\mathbf{g}_{goal,i}) \tag{8}$$

We complete the training procedure by optimizing the overall loss $\mathcal{L}_{total}$ as the weighted sum of the two losses in Eq. 9. We focus on improving the policy for performing the main task and assign weight $\eta$ to $\mathcal{L}_{GACE}$ for performing goal-aware representation learning for the feature extractor $\sigma(\cdot)$.

We note that the goal-discriminator $d(\cdot)$ is updated according to $\mathcal{L}_{GACE}$ only during training, not during inference, excluding the ActorCritic.

$$\mathcal{L}_{total} = \mathcal{L}_{RL} + \eta\mathcal{L}_{GACE} \tag{9}$$

The explained procedure forms a visual representation learning method, where the GACE loss makes the goal-discriminator become goal-aware without external supervision. Such goal-awareness is advantageous for sample-efficiency in multi-target environments, as well as generalization, as it makes the agent robust in a noisy environment. Such effects are demonstrated in our quantitative results and qualitative analyses.

### 3.4 Goal-discriminative attention networks

The goal-discriminator discussed so far can influence the policy inference. However, in order to effectively utilize the discriminator to enhance the performance and efficiency of the agent, we propose Goal-Discriminative Attention Networks (GDAN). Overall, the goal-aware attention described in Figure 1b involves a *goal-relevant query* $q_t^z$ from within the goal-discriminator $d(\cdot)$, and the *key* $k_t$ and *value* $v_t$ from encoded state in the ActorCritic $f(\cdot)$.

During the inference, the state encoding vector $\mathbf{e}_t$ is passed through the first linear layer of the goal-discriminator, which yields the query vector $q_t^z$ that implicitly represents the goal-relevant information. Meanwhile, the vector $\mathbf{e}_t$ is inferenced through a gated-attention [44] with instruction $I$ for grounding. The gated-attention vector is passed through LSTM (Long Short-Term Memory) [20] and splits in half into key $k_t$ and value $v_t$. Suppose the vectors $q_t^z$, $k_t$, and $v_t$ are $d_q$-, $d_k$-, and $d_v$-dimensional, respectively. The query and key are linearly projected to $d_v$-dimensional space using learnable parameters $\mathbf{W_q}$ of dimensions $d_v \times d_q$ and $\mathbf{W_k}$ of dimensions $d_v \times d_k$, respectively. The activation function $\phi$, which is $\tanh$ in our case, is applied to the resulting vectors to yield a goal-aware attention vector $u_t^z$ in Eq. 10. The attention vector $u_t^z$ contains the goal-relevant information in $s_t$. Lastly, Hadamard product is performed between $u_t^z$ and $v_t$ in Eq. 11, yielding the attention-dependent state representation vector $h_t$ used for calculating the policy and value. The vector $h_t$ is concatenated with gated-attention vector from a state encoding vector $\mathbf{e}_t$ in ActorCritic $f(\cdot)$.

$$u_t^z = \phi(\mathbf{W_q}q_t^z + \mathbf{W_k}k_t) \tag{10}$$
$$h_t = v_t \odot u_t^z \tag{11}$$

We underscore two points about the proposed networks. First, while the goal-discriminative feature extractor $\sigma(\cdot)$ trained by GACE loss may be sufficient, by constructing attention networks, the implicit goal-relevant information from the goal-discriminator directly affects the ActorCritic $f(\cdot)$ that determines the policy and value. Second, the query vectors are from a part of the goal-discriminator, which allows the ActorCritic to actively query the input for goal-relevant information rather than having to filter out large amounts of unnecessary information. These two design choices enable the agent to selectively allocate attention for goal-directed actions (Figure 5c), making full use of the GACE loss method. Details of our architecture are covered in Appendix B.

## 4 Experiments

**Experimental setup**  The experiments are conducted to evaluate the success ratio, sample-efficiency, and generalization of our method, as well as compare our work to competitive baseline methods in multi-target environments. We develop and conduct experiments on (1) visual navigation tasks based on ViZDoom [23, 18], and (2) robot arm manipulation tasks based on MuJoCo [42]. These multi-target tasks involve $N$ targets which are placed at positions $p_1, p_2, \cdots, p_N$, and provide the index of one target as the goal $z \in \{1, 2, \cdots, N\}$. At time step $t$, the agent receives state $s_t$ normalized to range [0,1]. The agent receives a positive reward $r_{success}$ if it gets sufficiently close to the goal (i.e. $\|x_t - p_z\| \leq \epsilon$) where $x_t$ is the agent position at time $t$. On the other hand, the agent is penalized by $r_{nongoal}$ if it reaches a non-goal object, or $r_{timeout}$ if it times out at $t \geq T$. Lastly, the agent receives a time step penalty $r_{step}$, which is empirically found to accelerate training. Reaching either a goal or non-goal object terminates the episode. All experiments are repeated five times.

**Details of sample efficiency metrics**  We introduce Sample Requirement Ratio (SRR) and Sample Efficiency Improvement (SEI) metrics to measure the sample efficiency of each algorithm. For the

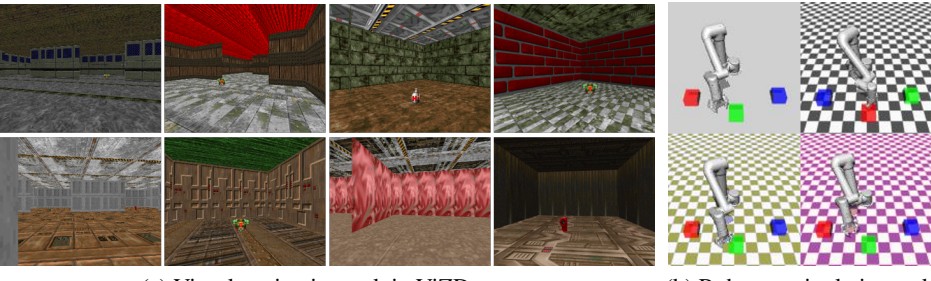

|  (a) Visual navigation task in ViZDoom  |  (b) Robot manipulation task  |

Figure 2: Two kinds of multi-target tasks in our experiments. The object positions and background are randomly shuffled in (a) egocentric visual navigation tasks and (b) robot arm manipulation tasks.

given task, we select a reference algorithm $A$ and its reference success ratio $X\%$. Suppose this algorithm reaches the success ratio $X\%$ within $n_A$ updates. We measure the SRR metric of another algorithm $B$ as $SRR_B = n_B/n_A$, where $B$ reaches $X\%$ within $n_B$ updates. Also, SEI metric of $B$ is measured as $SEI_B = (n_A - n_B)/n_B$. Lower SRR, higher SEI indicate higher sample efficiency.

### 4.1  Visual navigation task with discrete action

We conduct experiments on egocentric 3D navigation tasks based on ViZDoom (Figure 2a). The RGB-D state is provided as an input to the agent, and three different actions (*MoveForward*, *TurnLeft*, *TurnRight*) are allowed. One item from each of the four different classes of objects – each class (e.g. *Bonus*) containing two different items (e.g. *HealthBonus* and *ArmorBonus*) – is placed within the map, with one class randomly selected as the goal by the instruction for each episode like "Get the *Bonus*". In every episode, the agent is initially positioned at the center of the map. The rewards are set as $r_{success} = 10$, $r_{nongoal} = -1$, $r_{timeout} = -0.1$, $r_{step} = -0.01$. Full details of the environment are provided in the Appendix C.

**Details of each task**    To evaluate the performance of our method on tasks of varying difficulties, we set up four configurations. The **V1** setting consists of a closed fixed rectangular room with walls at the map boundaries, and object positions are randomized across the room. **V2** is identical to V1, except that the textures for the background, such as ceiling, walls, and floor, are randomly sampled from a pool of textures every episode. 40 textures are used for the *seen* environment, while 10 are used for the *unseen* environment which is not used for training to evaluate generalization. Hence the tasks allow $40^3$ and $10^3$ different state variations in *seen* and *unseen* environments respectively. **V3** is more complicated, with larger map size, shuffled object positions, and additional walls within the map boundaries to form a maze-like environment. "Shuffled positions" indicates that the object positions are chosen as a random permutation of a predetermined set of positions. Lastly, **V4** is equivalent to V3 with the addition of randomized background textures for the seen and unseen environments.V2 and V4 allow us to evaluate the agent's generalization to visually complex, unseen environments. V1 and V3 can be regarded as the upper bound of performance on V2 and V4 respectively.

**Baselines for comparison**    As the base algorithm, we use `A3C`, an on-policy RL algorithm, with the model architecture that uses gated-attention [5, 33] with LSTM. This baseline is proposed in [44][3] for multi-target learning on navigation. Other competitive general RL methods are added onto the base multi-target learning algorithm, due to the rarity of appropriate multi-target RL algorithms for comparison. `A3C+VAE` [16, 38] learns VAE features for all time steps, unlike our methods which learn only for goal states. `A3C+RAD` applies augmentation – the random crop method, known to be the most effective in [25] – to input states in order to improve sample efficiency and generalization. `A3C+GACE` and `A3C+GACE&GDAN` are our methods applied to the `A3C` as covered in Sec. 3.3 and 3.4.

**Results**    Performance is measured as the success ratio of the agent across 500 episodes in all tasks. As shown in Figure 4, most baselines successfully learn the V1 task, albeit at different ratios. In V4,

---

[3]The cited paper also introduces multi-target tasks, but those are unavailable due to license issuee.

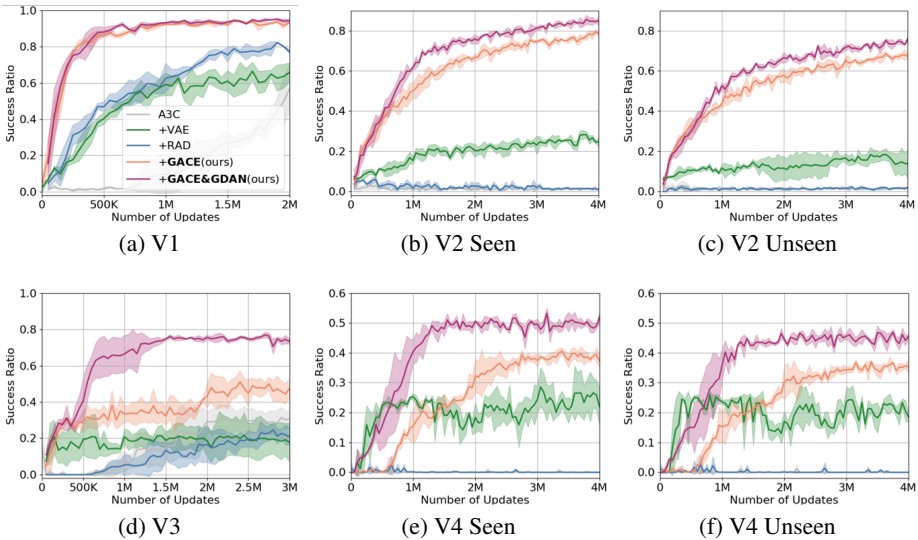

(a) V1          (b) V2 Seen          (c) V2 Unseen

(d) V3          (e) V4 Seen          (f) V4 Unseen

Figure 3: Visual navigation learning curves. The solid lines show the average success ratio over the repeated experiments, and the shades indicate bounds given as mean $\pm$ standard deviation of success ratio. Orange and red lines are our methods, GACE loss and GACE&GDAN, respectively.

`A3C+VAE` (green) initially shows the steepest curve, probably because it reaches the object located near the initial agent position before the learning commences properly. `A3C+RAD` (blue) shows the steepest learning curve in V1, but fails to learn in visually complex environments such as V2 and V4. In V3, `A3C+GACE` (orange) and `A3C+GACE&GDAN` (red) achieve 52.6% and 78.2% respectively. We attribute such discrepancy to `A3C+GACE&GDAN`'s more direct and efficient usage of goal-related information from the goal-discriminator. Especially, `A3C+GACE&GDAN` achieves as high as 86.6% and 54.9% in V2 seen and V4 seen respectively. Furthermore, in V2 unseen and V4 unseen tasks, the agents trained with our methods generalize well to unseen environments. Thus, our methods achieve the state-of-the-art sample-efficiency and performance in all tasks. The full details are covered in the Appendix B.

The SRR and SEI measurements of the models are shown in Table 1. For measuring sample efficiency, we select A3C as the reference algorithm and 56.55% as the reference success ratio in V1 task. This is the highest performance of A3C within 2M updates. Our methods show the highest sample efficiency compared to the baselines.

## 4.2 Robot manipulation task with continuous action

To evaluate our method in the continuous-action domain, we conduct experiments on the UR5 robot arm manipulation tasks with 6 degrees of freedom, based on MuJoCo (Figure 2b). The environment comprises a robot arm and three or five objects of different colors (red, green, blue, etc). The state is an RGB image provided from the fixed third-person perspective camera view that captures both the robot and the objects in Figure 2b. The rewards are $r_{success} = 1$, $r_{nongoal} = -0.3$, $r_{timeout} = -0.1$, $r_{step} = -0.01$.

**Details of each task** Our method is evaluated on three different vision-based tasks. In the **R1** task, the object positions are shuffled among three preset positions on grey background, and one of the three objects is randomly specified as the goal by the instruction like "Reach the *Green Box*". The robot arm must reach the goal object within the time limit $T$ steps while avoiding other objects. **R2** task complicates the R1 task by replacing the grey background with random checkered patterns. The agent's performance is measured on the *seen* environment, in which the agent is trained, as well as the *unseen* environment. The seen and unseen environments each uses a different set of 5 colors of the checkered background. **R3** task is a more complex variant of R1 setting, where five target classes are available, three of which are randomly sampled by the environment and randomly positioned among the three preset locations as in R1. The full details of these tasks are outlined in the Appendix C.

Table 1: Success ratio (SR) and sample efficiency metrics in visual navigation task **V1**. SRR (lower the better) and SEI (higher the better) are measured with A3C performance as a reference. "Number of Updates" indicates the number of updates required to reach the reference performance.

| Algorithm | SR of **V1** (%) | Number of Updates | SRR (%) | SEI (%) |
|---|---|---|---|---|
| A3C | | 2M | 100 | - |
| +VAE | | 810,086 | 40.50 | 146.89 |
| +RAD | 56.55 | 703,574 | 35.18 | 184.26 |
| **+GACE** (ours) | | 163,602 | 8.18 | 1122.48 |
| **+GACE & GDAN** (ours) | | 110,930 | **5.55** | **1702**.94 |

Table 2: Success ratio (SR) in robot arm manipulation tasks.

| Algorithm | SR of **R1** (%) | SR of **R2** Seen (%) | SR of **R2** Unseen (%) | SR of **R3** (%) |
|---|---|---|---|---|
| SAC | $63.1 \pm 6.9$ | $60.5 \pm 5.7$ | $53.4 \pm 6.9$ | $61.7 \pm 5.4$ |
| +AE | $67.2 \pm 5.0$ | $72.8 \pm 5.9$ | $59.4 \pm 5.5$ | $62.3 \pm 5.1$ |
| +CURL | $67.9 \pm 7.3$ | $74.5 \pm 9.2$ | $36.6 \pm 3.4$ | $64.7 \pm 4.0$ |
| **+GACE** | $84.7 \pm 10.0$ | $75.0 \pm 8.9$ | $63.0 \pm 9.0$ | $\mathbf{79.3 \pm 8.9}$ |
| **+GACE&GDAN** | $\mathbf{89.3 \pm 4.2}$ | $\mathbf{78.2 \pm 8.7}$ | $\mathbf{73.3 \pm 5.8}$ | $\mathbf{79.6 \pm 8.4}$ |

**Baselines for comparison**   As a baseline method, we use a deterministic variant of Soft Actor-Critic (`Pixel-SAC` [17]), an off-policy algorithm commonly used in robotic control tasks. As a baseline, we evaluate `SAC+AE` [45], which uses $\beta$-VAE [19] for encoding and decoding states. Another baseline is `SAC+CURL` which performs data augmentation (cropping) and contrastive learning between the same images. These baseline methods are all competitive methods in the continuous-action domain. We use SAC as a base algorithm to apply our methods `SAC+GACE` and `SAC+GACE&GDAN`. Note that the ActorCritic is as depicted in Figure 1b, without the LSTM.

**Results**   The results for robot manipulation tasks are shown in Table 2 and Figure 7 (in Appendix B). Performance is measured as the success ratio of the agent for 100 episodes. `SAC` shows higher difficulty in learning R2 tasks, which are more visually complicated than R1 task. Unexpectedly, `SAC+AE` and `SAC+CURL` has improved performance in R2 seen. We speculate that the two algorithms that perform representation learning are more suitable for the task with diversity. Especially, `SAC+CURL` consistently achieves competitive performances in all environments except for R2 unseen task. However, it does not seem capable of learning with complex unseen backgrounds, showing huge deterioration in performance for R2 unseen. `SAC+GACE` consistently shows strong performance in all environments, as well as high generalization capability for R2 unseen task. Finally, `SAC+GACE&GDAN` attains state-of-the-art performance in all environments including the unseen task for generalization. In particular, we observe the smallest performance discrepancy between R2 seen and unseen tasks. This shows that it can learn robustly in a complex background acting as a noise. Furthermore, our methods show the highest performance in R3 task, adapting well to a more diverse set of possible target objects. This supports that the GACE mechanism addresses the scalability issue well.

The SRR and SEI measurements of these methods are shown in Table 3. We select SAC as the reference algorithm and 63.1% as the reference success ratio in R1 task. `SAC+GACE` shows the state-of-the-art sample efficiency compared to all baseline models, meaning that it learns faster than `SAC+GACE&GDAN` for relatively easy tasks.

**Comparison with single-target task**   We conduct an ablation study, where effectively single-target policies are trained on a variant of R1 task with the instruction fixed to a single target as a goal. We train these policies for 120K updates, such that the total number of updates (360K) is roughly equal to the number of updates for training the multi-target baseline models. The average performance of these policies is 47±11%, which is noticeably lower than the performance of multi-target SAC agent (63.1±6.9%). Continuing the training, the single-target policies converge to 87.5±2.4% success ratio in 333,688 updates, serving as a competitive baseline. However, training individual single-target

Table 3: Sample efficiency metrics for R1 task. SR indicates the reference performance in R1 task.

| Algorithm | SR (%) | Number of Updates | SRR (%) | SEI (%) |
|---|---|---|---|---|
| SAC | | 314,797 | 100 | - |
| +AE | | 230,339 | 73.17 | 36.67 |
| +CURL | 63.1 | 142,480 | 45.26 | 120.94 |
| **+GACE** (ours) | | 53,774 | **17.08** | **485.41** |
| **+GACE&GDAN** (ours) | | 63,140 | 20.06 | 398.57 |

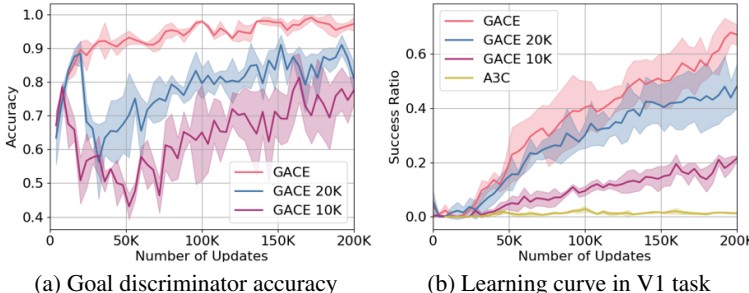

(a) Goal discriminator accuracy      (b) Learning curve in V1 task

Figure 4: Additional experiments to analyze the effectiveness of GACE loss. The accuracy of the goal-discriminator (a) and learning curve of the agent (b) are measured with the goal-discriminator weights unfrozen (red), frozen at 10K (purple) and 20K (blue) updates.

policies for multi-target tasks poses a scalability problem. It not only requires memory for weights that is directly proportional to the number of targets, but also uses a large amount of samples to learn the targets separately. In contrast, the multi-target agent can efficiently learn a joint representation, improving memory and sample efficiency as well as allowing generalizable learning of many targets.

### 4.3 Analyses

**Effectiveness of goal-aware cross-entropy loss**    Additional experiments are conducted in the V1 task to investigate how our method influences the agent's learning. To vary the extent to which the GACE loss may affect learning, the goal-discriminator weights are frozen after 10K and 20K updates such that the GACE loss $\mathcal{L}_{GACE}$ does not contribute to learning afterwards. We note that in Figure 4a, although the GACE loss (frozen weights) does not further contribute to learning, the discriminator accuracy improves only by updating the policy. This indicates that throughout the training, the agent gradually develops a feature extractor $\sigma(\cdot)$ that can discriminate targets. Our method makes it possible for the feature extractor to directly learn to discriminate.

In addition, the learning curves in Figure 4b corroborate that such development of the feature extractor is accelerated by the GACE loss. Even when the agent is trained with the GACE only temporarily (as with GACE 10K and 20K), the learning curve is steeper than that with vanilla A3C. Furthermore, the unfrozen GACE (red line) shows a steeper learning curve than GACE 20K, which is again steeper than GACE 10K. Consequently, it can be seen that representation learning of feature extractor updated by GACE loss has positive influence on policy performance than learning solely with policy updates.

**Simple ActorCritic** $f(\cdot)$    We perform an ablation study about attention model that simply concatenates $e_t$ from feature extractor $\sigma(\cdot)$ with $q_t^z$ from ActorCritic $f(\cdot)$ in Figure 1b to verify the effectiveness of the attention method. We conduct the experiment in R1 task. As a result, a success ratio of $81.5 \pm 10.0$ % is obtained, which is lower than that of GACE loss and GACE&GDAN. We speculate that since not every state corresponds to the goal – such goal states are quite sparse – the additional information of the goal-discriminator acts as a noise, hindering the efficient learning of the policy. This supports that, in contrast to such naive use of goal-discriminator features, our proposed attention method can effectively process goal-related information and is robust against noise.

**Analysis using saliency maps** To ascertain that an agent trained with GACE and GACE&GDAN indeed becomes goal-aware, we use saliency maps [15] to visualize the operation of three agents within the V2 unseen task, as shown in Figure 5. The blue shades indicate the regions that the agent significantly attends to for policy inference, and the red shades indicate those regions for the value estimation. The overlapping blue and red regions appear purple. The three agents are trained with A3C, GACE, and GACE&GDAN, respectively, for 4M updates.

Figure 5a shows the operation of the A3C agent. It shows overly high sensitivity to edges in the background, approaching the walls rather than searching for the goal. Also, in the rightmost image in (a), although it does attend to the goal object located on the right side, a *TurnLeft* action is performed. This suggests that the agent cannot discern between objects because it lacks understanding of targets. Figure 5b corresponds to the

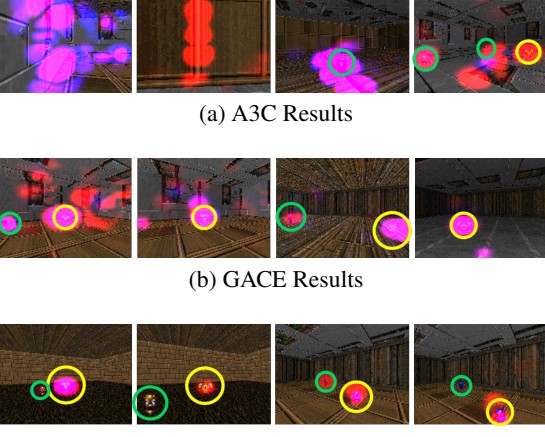

(a) A3C Results

(b) GACE Results

(c) GACE & GDAN Results

Figure 5: Visualization of saliency maps in V2 unseen (goal and non-goal in yellow and green circles respectively). (a) The agent is overly sensitive to edges in the background. (b) All goals and non-goals are detected successfully. (c) The agent shows sensitive reactions only to the goal.

GACE agent, showing high sensitivity to all targets and intermittent edges. As its attention to targets demonstrates, the ability to distinguish the goal is far superior to that of the A3C agent. Finally, Figure 5c shows that the GACE&GDAN agent exhibits high sensitivity to all goals while hardly responding to irrelevant edges. In addition, upon noticing the goal, the agent allocates attention only to the goal, rather than unnecessarily focusing on the non-goals. These results support that our method indeed promotes goal-directed behavior that is visually explainable.

## 5   Conclusion

We propose GACE loss and GDAN for learning goal states in a self-supervised manner using a reward signal and instruction, promoting a goal-focused behavior. Our methods achieve state-of-the-art sample-efficiency and generalization in two multi-target environments compared to previous methods that learn all states. We believe that the disparity between the performance improvements in the two benchmark suites is due to the degree of similarity between the goal states and the others.

The limitation of our method is that it may not be well-applied in tasks where instructions do not specify a target (e.g. "Walk forward as far as possible") or tasks with little correlation between states and rewards based on success signal. Also, our methods assume that the total number of target object classes is known beforehand. Our method can be abused depending on the goal specified by a user. Nonetheless, this paper brings to light the possible methods of prioritizing data for efficient training, especially in multi-target environment.

## Acknowledgments and Disclosure of Funding

We would like to thank Dong-Sig Han, Injune Hwang, Christina Baek for their helpful comments and discussion.
This work was partly supported by the IITP (2015-0-00310-SW.Star-Lab/10%, 2017-0-00162-HumanCare/10%, 2017-0-01772-VTT/10%, 2018-0-00622-RMI/10%, 2019-0-01371-BabyMind/10%, 2021-0-02068-AIHub/10%, 2021-0-01343-GSAI (SNU)/10%) grants, the KIAT (P0006720-ILIAS/10%) grant, and the NRF of Korea (2021R1A2C1010970/10%) grant funded by the Korean government, and the BMRR Center (UD190018ID/10%) funded by the DAPA and ADD.

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
