# APPENDIX

This appendix provides additional information not described in the main text due to the page limit. It contains additional analysis results in Section A, experiment details in Section B, environment details in Section C and a pseudocode of and SAC+GACE algorithms in Section D.

## A    Analysis

Our proposed methods show an outstanding performance in terms of sample efficiency, generalization, and task success ratio. This is consistent with the previous studies [38, 21], where auxiliary tasks for reinforcement learning encourage the agent to learn extra representations of the environment, as well as provide additional gradients for updates. To clarify more on the roles and characteristics of the proposed methods, we conduct an additional experiment.

### A.1    Generalization to unseen environment in visual navigation task

Figure 6 shows the accuracy curves of the goal-discriminator, or more precisely, how consistently the highest probability is assigned to the ground-truth label $z_i$ for the goal-discriminator output $\mathbf{g}_{goal.i}$. The goal state prediction accuracies of the goal-discriminator on V2 seen and V2 unseen tasks after 200K training updates are $\mathbf{99.32 \pm 0.62\%}$ and $\mathbf{89.15 \pm 3.44\%}$, respectively. The accuracy curves are shown in Figure 6. As the goal-discriminator learns various goal states, the agent becomes capable of discerning goals accurately even in an unseen environment. We believe this figure shows that GACE and GDAN can show high performance even in unseen tasks thanks to the excellent generalization ability of GACE.

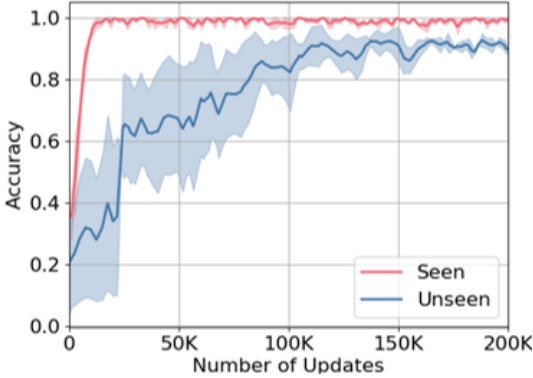

Figure 6: Goal prediction accuracy of goal-discriminator in V2 seen and unseen tasks.

## B    Architecture & experiment details

### B.1    Auto-labeled goal states for self-supervised learning

Prior to initiating the learning of goal-aware representation, the minimum amount of storage data is collected through a random policy during *warmup* phase. In the case of robot manipulation tasks, where a random agent achieves low success ratio, the states for the failed episodes are stored as a negative case $z_{N+1}$, where $N$ is the number of target objects in the environment. After the warmup phase, as the agent learns to succeed in the given task through trial and error, the successful goal states are stored. At the same time, due to their commonness, unsuccessful states – where the agent fails to reach any target object at all within time limit – are stored as negative case $z_{N+1}$ with a low probability $\epsilon_N < 1$. Empirically, setting $\epsilon_N$ value close to the success ratio of the random policy leads to high performance.

## B.2 Model for visual navigation task

The ViZDoom environment is a first-person perspective navigation task, requiring memory for reasonable inference. For this, we use a model that contains Long Short-Term Memory (LSTM), and all compared methods use the same model architecture.

Four consecutive frames of the environment are stacked into one state. The stacked state $s_t$ is fed as input to the image feature extractor $\sigma(\cdot)$, which consists of 4 convolutional layers, each with a kernel size of 3, a stride of 2, and zero-padding of 1. The first two layers each contains 32 channels and the rest of the layers each has 64 channels. The output of $\sigma(\cdot)$ is flattened and fed into a fully-connected layer of 256 units to be converted into image features $\mathbf{e}_t$. Batch normalization is applied after every convolutional layer. ReLU is used as the activation function for all of the layers.

We first describe the model corresponding to the GACE method without attention networks, and then elaborate the GDAN method afterwards. ActorCritic $f(\cdot)$ consists of a word embedding, an LSTM, an MLP for policy, and an MLP for value. The goal index from the instruction is converted to a word embedding $I$ of dimension 25. The embedding $I$ is fed into a linear layer and converted to $I'$ as a 256-dimensional vector. The gated attention vector $M$ is calculated as the Hadamard product between the image feature output $\mathbf{e}_t$ of the feature extractor and the instruction embedding $I'$.

$$I' = \text{Linear}(I) \tag{12}$$

$$M = \mathbf{e}_t \odot \text{sigmoid}(I') \tag{13}$$

The input to the LSTM is the concatenation between $M$ and $\text{sigmoid}(I')$. The hidden layer of the LSTM and the gated attention output $M$ are concatenated to form the input for both the MLP for policy and that for value. The MLP for policy consists of two layers with 128 and 64 units respectively and the MLP for value consists of two layers with 64 and 32 units respectively.

The goal-discriminator $d(\cdot)$ consists of MLP of two layers that respectively contains 256 and 4 units. Each of the four output units belongs to each target category. This goal-discriminator is only used during training.

GDAN adds an attention method to the architecture. At the end of the model, the attention-dependent state representation vector $h_t$, mentioned in Sec. 3.4 of the main paper, is concatenated with a state encoding vector $\mathbf{e}_t^1 := LSTM(\mathbf{e}_t)$. Lastly, the concatenated vector is fed as input to 2-layered perceptron (yet within $f(\cdot)$) to infer policy $\pi$ and value $V$.

## B.3 Success ratio comparison in visual navigation tasks

| Algorithm | **V1** (%) | **V2** Seen (%) | **V2** Unseen (%) |
|---|---|---|---|
| A3C | $56.55 \pm 13.75$ | $4.00 \pm 3.16$ | $3.99 \pm 2.30$ |
| +VAE | $67.89 \pm 3.50$ | $30.03 \pm 4.96$ | $19.79 \pm 3.06$ |
| +RAD | $82.14 \pm 2.34$ | $7.75 \pm 2.25$ | $3.87 \pm 2.78$ |
| **+GACE** (ours) | $\mathbf{94.97 \pm 0.70}$ | $79.52 \pm 0.83$ | $69.79 \pm 1.66$ |
| **+GACE&GDAN** (ours) | $\mathbf{95.63 \pm 0.64}$ | $\mathbf{86.62 \pm 1.48}$ | $\mathbf{76.36 \pm 1.53}$ |

Table 4: Success ratio comparison for V1, V2 tasks.

| Algorithm | **V3** (%) | **V4** Seen (%) | **V4** Unseen (%) |
|---|---|---|---|
| A3C | $33.45 \pm 4.72$ | $3.13 \pm 4.43$ | $4.40 \pm 6.22$ |
| +VAE | $26.26 \pm 2.17$ | $31.96 \pm 2.66$ | $28.18 \pm 0.27$ |
| +RAD | $27.91 \pm 3.48$ | $4.64 \pm 5.92$ | $9.98 \pm 7.65$ |
| **+GACE** (ours) | $52.58 \pm 4.09$ | $41.26 \pm 3.05$ | $37.96 \pm 1.61$ |
| **+GACE&GDAN** (ours) | $\mathbf{78.21 \pm 2.45}$ | $\mathbf{54.87 \pm 1.75}$ | $\mathbf{48.52 \pm 1.62}$ |

Table 5: Success ratio comparison for V3, V4 tasks.

### B.4   Model for robot arm manipulation task

Robot arm manipulation tasks are conducted in the MuJoCo environment. For these tasks, we construct a model shown in Figure 1b without LSTM and train the agent with Algorithm 1. Unlike the model for visual navigation tasks, we do not use LSTM within the model for robot arm manipulation tasks.

The main model architecture for the GACE method without attention in robot arm manipulation tasks is similar to the architecture used in visual navigation tasks, except that in place of LSTM, a single linear layer is used. At the end of the model, the attention-dependent state representation vector $h_t$ is concatenated with a state encoding vector $\mathbf{e}_t^1 := Linear(\mathbf{e}_t)$. The concatenated vector is used within the ActorCritic $f(\cdot)$ to infer policy $\pi$ and value $V$.

### B.5   UR5 manipulation task learning curves

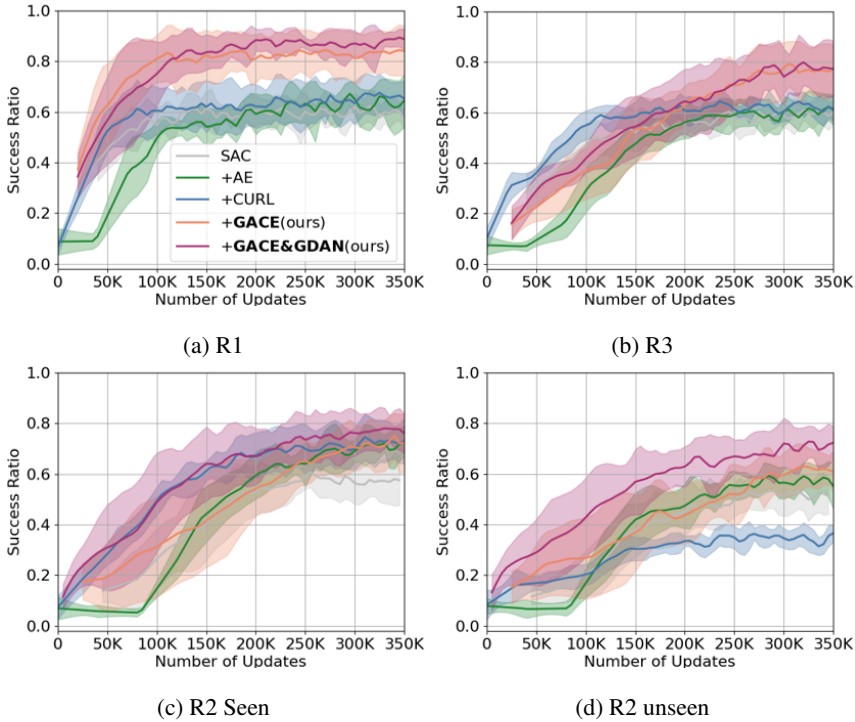

(a) R1                                 (b) R3

(c) R2 Seen                       (d) R2 unseen

Figure 7: UR5 manipulation task learning curves. The solid lines show the average success ratio over the repeated experiments, and the shades indicate bounds given as mean $\pm$ standard deviation of success ratio. Orange and red curves are our methods, GACE and GACE&GDAN, respectively.

### B.6   Hyperparameters

The hyperparameters for visual navigation tasks and UR5 manipulation tasks are shown in Table 6 and Table 7, respectively.

Most of the parameters commonly used in the base algorithm are used as in the correspondingly cited papers, and additional fine-tuning is conducted to suit the learning environment. In addition to the parameters suggested by our method, a value of [0.3, 0.5] is recommended for $\eta$, and it is recommended to set $\epsilon_N$ close to the success ratio of the random policy.

## C   Environment details

The additional details regarding the visual navigation tasks in ViZDoom and the robot arm manipulation tasks in MuJoCo that could not be covered in the main paper are outlined in this section.

## C.1   Visual navigation in ViZDoom

Each visual navigation task provides RGB-D images with dimensions $42 \times 42$. The images of four consecutive frames are stacked into one input state $s_t$. In **V1** and **V2** tasks, the maximum number of time steps in an episode is $T = 25$, and the size of maps for these tasks is $7 \times 7$. All objects are randomly located. The success ratio of random policy is $\mathbf{6.6}\%$. In **V3** and **V4** tasks, the maximum number of time steps in an episode is $T = 50$, and the size of maps for these tasks is $10 \times 10$. All object positions are shuffled, such that the object positions are chosen as a random permutation of a predetermined set of positions which are chosen evenly across the map. The success ratio of random policy is $\mathbf{8}\%$. An action repeat of 4 frames is used.

## C.2   Robot manipulation task in MuJoCo

Input state $s_t$ is an RGB image with dimensions of $84 \times 84$. No frame stack is used. In each time step, an action is repeated 16 times. The episode time limit is $T = 50$ steps in all tasks. The joint angles of the robot are constrained within specific boundaries, which are [-3.14, 3.14] or [-5, 5]. For the R2 tasks, the background in each task is randomly sampled from a set of five checkered patterns, some of which are shown in Figure 8. The generalization ability of the agent is measured by making the set of backgrounds in the R2 seen task and the set of backgrounds in the R2 unseen task mutually exclusive. The generalization ability of the agent is measured by assigning five background checkered colors to each of the seen and unseen tasks.

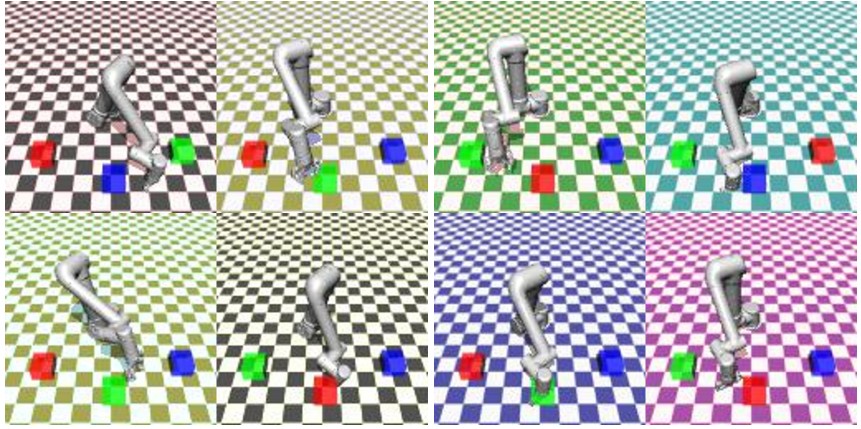

(a) Samples of R2 seen environments    (b) Samples of R2 unseen environments

Figure 8: Examples of observations in R2 seen and unseen tasks.

| Parameter Name | Value |
| --- | --- |
| Warmup | 2,000 |
| Batch Size for Goal-discriminator | 50 |
| GACE Loss Coefficient $\eta$ | 0.5 |
| Negative Sampling $\epsilon_N$ | 0 |
| Discount $\gamma$ | 0.99 |
| Optimizer | Adam |
| AMSgrad | True |
| Learning Rate | 7e-5 |
| Clip Gradient Norm | 10.0 |
| Entropy Coefficient | 0.01 |
| Number of Training Processes | 20 |
| Back-propagation Through Time | End of Episode |
| Non-linearity | ReLU |

Table 6: Hyperparameters used in visual navigation experiment.

| Parameter Name | Value |
| --- | --- |
| Warmup | 10,000 |
| Negative Sampling $\epsilon_N$ | 5% |
| Tau | 0.005 |
| Batch Size for SAC | 128 |
| Batch Size for Goal-discriminator | 128 |
| Hidden Vector Size | 256 |
| Target Update Interval | 1 |
| Replay Buffer Size | 1,000,000 |
| Goal Storage Size | 500,000 |
| GACE Loss Coefficient $\eta$ | 0.5 |
| Hidden units | 256 |
| Discount $\gamma$ | 0.99 |
| Optimizer | Adam |
| Learning Rate | 7e-5 |
| Entropy Coefficient | 0.01 |
| Non-linearity | ReLU |
| Optimizer | Adam |

Table 7: Hyperparameters used in robot arm manipulation experiment.

# D Algorithm

---
**Algorithm 1** SAC + GACE

---
Initialize SAC parameters
Goal-discriminator parameters: $\theta_g$
Policy parameters: $\phi$
Replay buffer $\mathcal{D} \leftarrow \emptyset$, Goal-discriminator Data storage $\mathcal{D}_g \leftarrow \emptyset$
**for** each iteration **do**
   **for** each environment step **do**
      Get state $s_t$, instruction $I^z$ and perform $a_t \sim \pi_\phi(a_t|s_t, I^z)$
      Get reward $r_t$ and next state $s_{t+1}$
      $\mathcal{D} \leftarrow \mathcal{D} \cup \{(s_t, a_t, r_t, s_{t+1})\}$
      If Success, $\mathcal{D}_g \leftarrow \mathcal{D}_g \cup \{(s_t, one\_hot(z))\}$
      If Fail, $\mathcal{D}_g \leftarrow \mathcal{D}_g \cup \{(s_t, one\_hot(z_{N+1}))\}$ with probability $\epsilon_N$
   **end for**
   **for** each gradient step **do**
      SAC policy update
      $\theta_g \leftarrow \theta_g + \eta\nabla_{\theta_g}\mathbb{E}_{(s,z)\sim\mathcal{D}_g}[one\_hot(z) \cdot \log(d(\sigma(s)))]$
   **end for**
**end for**

---