# OpenReview forum: "Goal-Aware Cross-Entropy for Multi-Target Reinforcement Learning"
_NeurIPS.cc/2021/Conference — NeurIPS 2021 Poster_

### Official Review · Reviewer_PmsL · 2021-07-14

**Rating:** 6
**Confidence:** 3

**Summary:**

This paper presents a method for training multi-target policies using reinforcement learning. It introduces an auxiliary classification-type loss for a feature encoder on which RL is performed, and additionally, uses an attention-based architecture to compute the policy and value function. In combination, these two elements seek to learn features which can distinguish between different goals in the environment, and then have the RL algorithm directly use these features to compute goal-relevant policy outputs. The authors show that adding these two components to existing RL algorithms improves performance and sample efficiency on simulated visual navigation and robotic manipulation environments compared to the base versions of these algorithms.

**Limitations And Societal Impact:**

As pointed out in the conclusion of the paper, there is some disparity between the performance on the two tasks, and the authors note that this is due to the degree of similarity between goal states and others (assuming it is meant the similarity between goal states and other states in the manipulation setting). I agree with this assessment. The authors also address an important limitation that tasks must specify targets, and have high reward-state correlation.

**Main Review:**

Originality: Taking features from the learned goal-discriminator and using it as an attention input is novel to me and seems quite effective. The overall method is able to demonstrate impressive performance on previously unseen visual navigation settings, likely through attending to only goal-relevant portions of the image observations.

Quality: The method seems to provide significant improvements over the base algorithms in the experiments demonstrated. The qualitative results (Figure 5) are compelling, supporting the authors’ arguments.

One point which could be investigated a bit more thoroughly: the GACE loss function appears to be analogous to training a classifier on goal images, which is similar to methods such as CURL but assumes that one can identify goal states for multiple tasks, which CURL does not use. If contrastive learning were performed on goal images instead, would this result in better performance than the SAC+CURL baseline presented?

Clarity:
Mostly, section 3.2 is confusing, discussing auto-labeled goal states -- how are the reached goal label z and corresponding goal state automatically collected during training? It seems the way in which the goal state is identified is through the last timestep of the episode $t^\prime$. But the termination condition seems to come from the environment, which then provides supervision. If this interpretation is correct, I feel that this is not quite completely self-supervised learning, and rather a multi-task sparse reward setting -- the environment still needs to provide the label for when the task is completed.

It would also be interesting to see how much using goal-discriminator features in the attention mechanism improves performance. What if the outputs of $\sigma$ are simply used instead?

Another useful comparison point would be the performance of training separate policies on each task (e.g., use a different actor/critic each time z changes). Because the proposed method is not goal-conditioned but rather instruction-conditioned, and so has a fixed set of possible instructions, this could be a good baseline to compare to.

Nitpick: Is the description on L138 necessarily true? This seems to depend on the particular rewards chosen for each task.

The paper is generally well-written, although as mentioned above, Section 3.2 was slightly confusing to me.

Significance:
Using GACE+GDAN leads to vastly better sample efficiency for the navigation task, even in settings where the baseline algorithms can solve the task.

The experiments on the continuous robotic control domain are a bit weaker than the visual navigation task -- there are only three possible object positions, and the tasks are pushing and reaching, so all algorithms produce relatively strong performance. It seems that SAC+CURL is able to achieve almost the same performance (within 1 standard deviation) of GACE+GDAN, with very similar sample efficiency. Therefore it is hard to say that the method is improving much here. An even more complex manipulation task could be insightful.

Recommendation: The ideas presented in this paper are quite nice and it also has some impressive empirical results. However, the robotic manipulation environments don't seem to significantly improve, the method could use additional comparisons as mentioned above, and it may not be suitable to tackle completely self-supervised settings as the authors claim. Therefore I feel that in its current state the paper should not yet be accepted.

**Time Spent Reviewing:**

3.5

---

> ### Author Response · Authors · 2021-08-10
> **Response to Reviewer PmsL**
>
> Thank you very much for your insightful feedback. We provide detailed responses below, including the additional experiments we newly conducted according to your suggestions.
>
> **How are the reached goal label z and corresponding goal state automatically collected during training? [...] But the termination condition seems to come from the environment, which then provides supervision. If this interpretation is correct, I feel that this is not quite completely self-supervised learning, and rather a multi-task sparse reward setting -- the environment still needs to provide the label for when the task is completed.**
>
> For reinforcement learning, we used automatic labeling of data without additional external information other than trajectories including state and reward, which are basically obtained through interactions with the environment. Rather than supervision, the automatic labeling is done from context only with such trajectory information and instructions, so it can be considered as a work that matches the definition of self-supervised learning. Specifically, our proposed method can be viewed as a self-supervised learning approach that performs self-supervision using a reward signal and instruction.
>
> **It would also be interesting to see how much using goal-discriminator features in the attention mechanism improves performance. What if the outputs of $\sigma$ are simply used instead?**
>
> This sounds like a fascinating suggestion, so we conducted an additional experiment in R1 by simply concatenating the output $q_t^z$ of the goal-discriminator and outputs of $\sigma$ as an ablation study of the attention model. As a result, a success ratio of 85.2 $\pm$ 3.9 \% was obtained, which is significantly lower than that of GACE (93.0 $\pm$ 0.7 \%) and GACE&GDAN (95.5 $\pm$ 1.5 \%). We speculate that since not every state corresponds to the goal -- such goal states are quite sparse -- the additional information of the goal-discriminator acts as a noise, hindering the efficient learning of the policy. This supports that, in contrast to such naive use of goal-discriminator features, our proposed attention method can effectively process goal-related information and is robust against noise.
>
> **Another useful comparison point would be the performance of training separate policies on each task (e.g., use a different actor/critic each time z changes).**
>
> We performed new experiments where effectively single-target policies are trained. In detail, three policies were trained on R1 task while fixing the instruction to a single target, practically making this a single-target task for each policy. We trained these policies for 120K updates each, such that the total number of updates (360K) roughly matches the number of updates for training the baseline multi-target agents (350K). As a result, the single-target policies achieved a success ratio of 77.5 $\pm$ 9.8\% on average, which is comparable to the performance of multi-target SAC policy (74.0 $\pm$ 7.2\%). Continuing the training, these policies converge to a maximum success ratio of 89.1 $\pm$ 3.1\% in 170K updates, which we can use as a competitive baseline.
>
> It should be warned, nonetheless, that using many separate policies has a scalability issue. It requires memory directly proportional to the number of targets, and larger amount of data is needed for convergence (each policy required roughly 170K updates on average, which adds to a total of 510K updates, which is approximately 60\% more samples than those required by SAC for converging).
> On the other hand, the proposed method learns representation that applies over multiple targets, allowing more sample-efficient and generalizable learning of many targets.
>
> **The experiments on the continuous robotic control domain are a bit weaker than the visual navigation task [...] Therefore it is hard to say that the method is improving much here. An even more complex manipulation task could be insightful.**
>
> Following your suggestion, we conducted additional experiments on a more complex variant of R1 setting, where five target classes are available, three of which are randomly sampled by the environment and randomly positioned among the three preset locations as in R1. The success ratios of the baseline methods and our methods (measured up to 450k updates) are as below:
> - SAC: 57.5 $\pm$ 17.3
> - SAC+AE: 66.1 $\pm$ 3.7
> - SAC+CURL: 70.7 $\pm$ 4.6
> - SAC+GACE (ours): 74.0 $\pm$ 9.3
> - SAC+GACE&GDAN (ours): $\mathbf{80.4} \pm \mathbf{6.9}$
>
> The discrepancy between SAC+GACE&GDAN and SAC+CURL is more noticeable in this more complex setting, suggesting that our method may be scalable to problems of higher difficulty.

---

> > ### Comment · Reviewer_PmsL · 2021-08-24
> > **Thank you for the response**
> >
> > Thank you for the thorough responses, clarification, and additional experimentation! I still do not quite understand the discussion about self-supervision, since self-supervised RL generally refers to methods which do not use environment rewards. I may be misunderstanding what the term is meant to imply there.
> >
> > The experiments with the goal-discriminator features are quite interesting and I agree that the authors' explanation seems feasible. The additional baseline with separate policies for each task are also very helpful -- they give a good sense of task difficulty and I agree with the authors that lack of scalability makes this naive strategy often impractical. Finally, the results on the more complex variant of R1 is more convincing as well.
> >
> > I will adjust my score to a 6 since I feel that the concerns about the comparisons and experiments I had before have been mostly resolved.

---

> > > ### Author Response · Authors · 2021-09-13
> > > **Thanks to Your Response and Insightful Comments**
> > >
> > > We are glad that most of your concerns about the comparison and experiments have been addressed.
> > > Again, we highly appreciate your suggestions for new experiments, as these provide further insights and strengthen our paper.
> > >
> > > Best, Authors.

---

### Official Review · Reviewer_LzLd · 2021-07-14

**Rating:** 6
**Confidence:** 3

**Summary:**

This paper introduces a novel approach for improving sample efficiency in what they call "multi-target reinforcement learning". This is essentially goal-conditioned reinforcement learning where the tasks are of the form "bring me a spoon" or "go to the kitchen". Sample efficiency is improved in two ways. Firstly, they introduce a new auxiliary task of training a goal-discriminator that takes as input a state embedding and predicts which goal this state achieves. Second, they introduce a new architecture that uses attention and information from the goal discriminator to better extract task-relevant features for the policy.

Several experiments are provided to test the sample efficiency of agents trained using the method in both visual navigation tasks and robotic manipulation tasks. In both settings, their method outperforms on-policy baselines. Additional analysis is done, showing that: (1) while explicitly optimizing the state representation to improve the goal discriminator results in the fastest improvement in accuracy in the discriminator, the RL loss itself still causes the representations to contain information about the goal, just slower. This shows that the representation learning loss is helping the model learn faster what it would have to learn anyways. And finally, (2) saliency maps are provided which show that when using attention, the policy attends to more interpretable regions of the state space.

**Ethical Concerns:**

I have no ethical concerns with this paper.

**Limitations And Societal Impact:**

The paper does a good job of listing some potential limitations of their work.

**Main Review:**

1. Originality: While representation learning and goal-conditioned RL are popular these days, the idea of using pairs of observations and the tasks that they solve to improve learned representations and reduce sample complexity seems novel.
2. Quality: I think this is a neat idea that makes better use of the assumptions we already make when we do goal-conditioned RL (like there being some idea of "targets" that we have access to during training). I had a few questions when reading this that I am curious about:
    1. Is there a reason why on-policy RL is used rather than off-policy + HER? If a practitioner is looking for a sample-efficient algorithm for goal-conditioned RL, they will likely use some combination of off-policy RL and HER. It would be great to see whether this representation learning approach works well with those algorithms.
    2. In Section 3.2, you describe the method for collecting $s_t$ and $z$ pairs. This seems like a bit of a chicken and egg problem, where with a random policy I would imagine that there would be a very low chance of actually achieving a goal and getting a positive example, but it seems positive examples are needed to learn a good representation to learn the task. It seems like this would be trivial if you made the assumptions made in HER (that you can get a goal label for any state during training in order to relabel unsuccessful trajectories). Is this something you considered? Is there a reason you do not assume access to something like this since it seems necessary in order to evaluate the reward anyways.
3. Clarity: I think clarity is the main area where this paper is lacking. Some points:
    1. The introduction does not do a great job of motivating the approach or describing it.
    2. Section 3.3 for the most part makes sense (you are classifying states based on their achieved goal), but the motivation is not clearly described. For example, on line 137 you write "[the] vanilla base algorithm is not sufficient for gaining direct understanding of goals". What does it mean for an algorithm to be not sufficient at this task? What does gaining a "direct understanding of goals" mean? It seems like since these multi-target RL tasks are just a collection of RL tasks, that an algorithm like A3C should be able to eventually solve them?
    3. Section 3.4 is very unclear and I am still confused about GDAN after thoroughly reading the paper. I think this needs to be set up with much better motivation and described in a more intuitive way.
    4. In Section 4.3, I am wondering how the discriminator accuracy goes up when it is not being trained anymore? This result would make sense if the discriminator was still learning but just not passing gradients back to the state encoder, but this isn't how it is described.
4. Significance: This type of representation learning approach seems applicable in many applications and seems simple to implement (at least for GACE).

Overall, I think the ideas behind the paper are solid (at least for GACE, I still need more clarity on exactly what is happening with GDAN). I would like to see the writing cleaned up and better explanations for some parts listed above.

**Time Spent Reviewing:**

4

---

> ### Author Response · Authors · 2021-08-10
> **Reponse to Reviewer LzLd**
>
> We highly appreciate your helpful feedback and concerns. Overall, we will reflect your comments in the final submission of our paper to improve clarity and motivation. We provide our detailed responses to your questions as below.
>
> **Is there a reason why on-policy RL is used rather than off-policy + HER?**
>
> We would like to point out that SAC (used for robot arm manipulation task) is an off-policy RL method, while A3C (used for navigation task) is on-policy. Thus, GACE and GDAN can be applied to off- and on-policy RL methods alike.
> HER is suitable for single-target tasks where it is easy to determine whether a goal has been reached when an episode fails. However, in order to utilize HER in multi-target environments, we must assume that it is possible to know which goal has been reached even when it fails. For example, suppose that the environment has targets A, B, and C, and the agent reaches B when the goal is A. Without the assumption, the agent cannot know which of B or C has been reached (or not reached) just with the fact that it did not receive a success reward. Therefore, it is difficult to apply HER in multi-target environments. However, since our method only needs to use the success signal that is basically given in a reinforcement learning environment, there is no need for such an additional signal.
>
> **In Section 3.2, [...] It seems like this would be trivial if you made the assumptions made in HER (that you can get a goal label for any state during training in order to relabel unsuccessful trajectories).**
>
> We think HER may be a suitable approach given that such assumptions are satisfied. However, it can only be applied effectively if the experienced states correspond to some target. The additional assumption (that the goal label can be obtained for any state) is not likely to be helpful, since the states in which the agent interacts with any target at all are quite sparse. Moreover, the environments that satisfy such assumptions are very rare, which limits the scope of application. In effect, we propose a method that assumes a more general and realistic environment.
>
> **What does it mean for an algorithm to be not sufficient at this task? What does gaining a "direct understanding of goals" mean?**
>
> In our study, the agent can be considered to have a "direct understanding of goals" because the goal discriminator directly discriminates the goal state through GACE, and the feature extractor directly learns about the goal. On the other hand, the vanilla base algorithm does not have such a mechanism, but it can eventually learn to distinguish goals to some extent. It would have been clearer to phrase that the vanilla base algorithm is “inefficient/indirect” at gaining a direct understanding of goals, rather than being “not sufficient”. We will reflect this in the final submission of the paper.
>
> **Section 3.4 [...] needs to be set up with much better motivation and described in a more intuitive way.**
>
> The motivation behind GDAN has been discussed in the Introduction (lines 32-39, lines 47-48), and the design choices and the intuition for GDAN have been covered in Section 3.4 (lines 178-185). As an additional note, with only the GACE loss/goal-discriminator, the goal-relevant knowledge learned with GACE loss affects the policy (or more precisely, ActorCritic $f(\cdot)$) only through the feature extractor $\sigma(\cdot)$. We figured that this was quite indirect, as the goal-discriminator weights would not contribute to the policy. Hence, we connected the goal-discriminator to the ActorCritic module $f(\cdot)$ via an attention mechanism in order to utilize the goal discriminator's information more directly in policy. We will rearrange the wordings to emphasize the motivation in the final submission of our paper.
>
> **Related to Section 4.3, [...] how the discriminator accuracy goes up when it is not being trained anymore?**
>
> After the weights of goal-discriminator $d(\cdot)$ are frozen, the goal-discriminator stops learning, and the gradients of the GACE loss $L_{GACE}$ are not propagated. Nevertheless, the weights of the feature extractor $\sigma(\cdot)$ are not frozen, and the agent continues to adjust these weights according to the RL loss. Thus, the increase in the goal-discriminator performance is attributed to the learning of the feature extractor. While this setup and conclusion is as explained in the lines 287-292, we will revise our paper for better clarity.

---

> > ### Comment · Reviewer_LzLd · 2021-08-26
> > **Response**
> >
> > After reading through the responses, I have a better understanding of the paper. Assuming that changes are made to the writing in the paper, my concerns and confusion have been largely addressed and I feel comfortable changing my score to a 6.

---

> > > ### Author Response · Authors · 2021-09-13
> > > **Thanks to Your Response and Questions**
> > >
> > > We are delighted to hear that your concerns have been addressed.
> > > Thank you again for your valuable comments.
> > >
> > > Sincerely yours, Authors.

---

### Official Review · Reviewer_TeEW · 2021-07-16

**Rating:** 7
**Confidence:** 3

**Summary:**

This paper proposes an auxiliary loss function and a model architecture to facilitate RL agent’s semantic understanding. Specifically, a separate discriminator that partially shares weights with the policy is trained to select the correct goal given the input instruction, while another attention module is trained to better extract the goal-relevant features. Experiments show that the proposed method improves both the sample efficiency and generalization dramatically on instruction-following tasks in visual navigation and robot manipulation.


**Limitations And Societal Impact:**

Yes

**Main Review:**

*Originality*: The method proposed by this paper is simple and sound. The related works are adequately cited so the contribution is clear.

*Quality*: I believe the proposed method is technically sound and used appropriately. The work is complete and both major contributions are backed up by clean ablation study. The authors are honest about their weaknesses too (e.g., it may not be well-applied to a task where the goal is not associated with a target).

However, I wonder whether it’s true that the target also needs to be known beforehand? For example, one needs to first define a fixed set of targets that potentially can show up in the environment before training the agent. If this is true, then I think authors should add this as another weakness as it’s hard to achieve in an open-world.


*Clarity*: The paper is pretty well-written and easy to understand. The analyses in section 4.3 are very informative.

*Significance*: I think the results are cool and improves the state of the art in a demonstrable way. Additionally, other researchers will likely build on this approach to learn better representation for RL agents because it’s easy to implement.


**Time Spent Reviewing:**

2.5

---

> ### Author Response · Authors · 2021-08-10
> **Response to Reviewer TeEW**
>
> We sincerely thank you for your feedback and support of our paper submission. We address your comments below.
>
> **The target also needs to be known beforehand?**
>
> Rather than specifying all the targets in advance, our method simply operates based on the instruction and success signals provided by the environment during the learning process. In this paper, the agents were not previously informed which targets exist in the environment. Nonetheless, in calculating the GACE loss, our method does assume that the total number of target object classes is known beforehand. A promising direction of future work, as we mention in the reply to the second question of Reviewer RRiW, is to interact with dynamic targets that are specified by target images as a cue (along with the works of scene-driven navigation [1,2,3]).

---

> > ### Comment · Reviewer_TeEW · 2021-08-24
> > **Thanks for the responses**
> >
> > Got it, the clarification is clear. I believe adding this assumption into the paper will help readers to understand.

---

> > > ### Author Response · Authors · 2021-09-13
> > > **Thanks to Your Response and Support**
> > >
> > > We will revise the final draft to account for the assumption. Thank you.
> > >
> > > Regards, Authors.

---

### Official Review · Reviewer_RRiW · 2021-07-28

**Rating:** 7
**Confidence:** 4

**Summary:**

The paper addresses problem of sample efficiency and generalization in the multi-target reinforcement learning. The author proposes goal-aware cross-entropy (GACE) loss for auto-labeling goal states. The author also develops goal-discriminative attention network (GDAN) for multi-target reinforcement learning. They evaluate the proposed method on visual navigation robot arm manipulation with multi-target environment.  The proposed method in the paper compares with A3C, SAC.

**Limitations And Societal Impact:**

- Could you please explain RS-D2 in line 18.
- The multi-target setup has also a challenge of target changing or dynamic targets, could you talk about how the method would perform in dynamic targets situations?
- In section 5, the paper talks about one of the limitation to the proposed method as target ambiguity. Could you please elaborate a little more?
- I understand that with the pandemic situation it would be difficult to access lab to conduct real-world experiments. If possible could the authors talk about how the methods can be extended to real-world robotics tasks?

**Main Review:**

- The sample efficiency and generalization for multi-goal RL are interesting problems to be addressed.
- The proposed GACE loss and GDAN method are nice contributions for research community.
- The paper evaluates good range of experiments.
- The proposed method in the paper achieves strong results against the baselines.
- The paper is well written.

**Time Spent Reviewing:**

7

---

> ### Author Response · Authors · 2021-08-10
> **Response to Reviewer RRiW**
>
> Thank you very much for your feedback and time spent reviewing our work. We address your comments below.
>
> **Could you please explain R2-D2 in line 18.**
>
> The R2-D2 robot in line 18 is an artificial intelligence robot that appears in the movie Star Wars. While a lighthearted reference, we mentioned it as a popular example of a robot with human-level intelligence that plays a role in helping people. In order to reach that level, the issues addressed by our paper can be a cornerstone.
>
> **The multi-target setup has also a challenge of target changing or dynamic targets, could you talk about how the method would perform in dynamic targets situations?**
>
> Depending on what was meant by “target changing or dynamic targets”, the approach may be somewhat different.
> 1) If each target can take various or changing appearances (e.g. the task involves a “ball” target of various colors or a “toy ambulance” target that constantly flashes), we can apply our method as is, similarly to the settings of our visual navigation benchmark.
> 2) In cases where the number of possible targets gradually increases throughout the training, we may be able to address the problem by extending our proposed method with incremental learning methods for multi-class classification.
> 3) In situations where the agent must be able to reach or interact with any previously unseen targets at test time, there are approaches that use images of target objects that are provided as cues (scene-driven navigations [1,2,3]). While this is outside the scope of our research in this paper, we think it is a challenging issue worth considering for future research.
>
> **In section 5, the paper talks about one of the limitation to the proposed method as target ambiguity. Could you please elaborate a little more?**
>
> For example, if the instruction is provided as "Walk forward as far as possible", the aim of the task does not involve reaching a specific target destination or interacting with a specific target object. In tasks such as Half-Cheetah, Walker-2d, and Ant, there may not even exist target objects to interact with. Consequently, our method is not applicable to these tasks.
>
> In lines 337-339, we also mentioned "tasks with little correlation between the states and rewards based on success signals." Specifically, the environment may be designed such that successfully reaching/interacting with the goal object does not necessarily lead to high reward, or there may even be misleading rewards that distract the agent away from the actual goal object.
>
> **If possible could the authors talk about how the methods can be extended to real-world robotics tasks?**
>
> Robot exploration in the real world raises additional problems, such as posing safety concerns and constraining the agent to operate in real time. Hence, it is imperative to avoid random exploration in the real world as much as possible. These issues can be addressed by training the agent in a safety-constrained environment, or by pretraining the policy via offline RL or simulation. Afterwards, the proposed GACE/GDAN method can be applied by establishing a real-world environment that can give reward signals through the sensor.
>
> [1] Devo, A., Mezzetti, G., Costante, G., Fravolini, M. L., and Valigi, P. (2020). Towards generalization in target-driven visual navigation by using deep reinforcement learning. IEEE Transactions on Robotics, 36(5):1546–1561.
>
> [2] Mousavian, A., Toshev, A., Fišer, M., Košecká, J., Wahid, A., and Davidson, J. (2019). Visual representations for semantic target driven navigation. In 2019 International Conference on Robotics and Automation (ICRA), pages 8846–8852. IEEE.
>
> [3] Zhu, Y., Mottaghi, R., Kolve, E., Lim, J. J., Gupta, A., Fei-Fei, L., and Farhadi, A. (2017).
> Target-driven visual navigation in indoor scenes using deep reinforcement learning. In IEEE
> international conference on robotics and automation, pages 3357–3364.

---

### Decision · Program_Chairs · 2021-09-27

**Decision:**

Accept (Poster)

**Comment:**

All reviewers agree that the multi-target RL settings, where the agent is to reach specified and variable goals (called targets) depending on the task specification, is an important setup and the proposed extension of RL formulation -- a loss to predict goal states and a second to focus on the features relevant for the goal -- are novel, well motivated, and empirically supported. Some of the reviewers recommend to slightly improve the writing and further clarify the self-supervision setup, which we believe can be easy accomplished for the final version. Hence accept.